# Probing weak chemical interactions of metal surface atoms with CO-terminated AFM tips identifies molecular adsorption sites

Jalmar Tschakert[1,2], Qigang Zhong [1,2], Alexander Sekels[1,2], Pascal Henkel [2,3,4], Jannis Jung[2,3], K. Linus H. Pohl[2,3], Hermann A. Wegner [2,5], Doreen Mollenhauer [2,3,6,7,8] ✉, André Schirmeisen [1,2] & Daniel Ebeling [1,2] ✉

Metal surfaces play a key role in on-surface synthesis as they provide a two-dimensional catalytic reaction environment that stimulates activation, diffusion, and coupling of molecular reactants. Fundamental understanding of the interactions between surface atoms and reactants is very limited but would enable controlling on-surface reaction processes for designing functional nanomaterials. Here, we measure chemical interactions between CO-terminated tips and Cu(111), Ag(111), and Au(111) surface atoms in all spatial directions with picometer resolution via low temperature atomic force microscopy. This allows a site-specific quantification of the weak chemical interactions of densely packed metal surface atoms and provides a picture of the potential energy landscape experienced by adsorbed reactants. Accompanying density functional theory calculations and the crystal orbital overlap population method reveal small covalent binding contributions from orbital overlap of the corresponding p- and d-states of the CO tip and the metal surface atoms as the cause for the site-specific interactions. Accessing such small covalent bonding contributions in the background of the dispersion-dominated interaction enables revealing insights into the nature of chemical bond formation with metal surface atoms and a reliable determination of molecular adsorption sites. The latter can serve both as a starting point and as a direct comparison with theoretical studies.

The ability to construct complex organic nanomaterials with atomic precision via the so-called on-surface synthesis[1–3] has recently led to a plethora of functional materials that may be used as tunable light emitters[4], flexible molecular-scale circuitry[5–11], spintronic devices[12–15], for probing quantum effects[16,17], or in field effect transistors[18,19]. However, creating such materials is not straightforward since the field

of on-surface synthesis is relatively new and the underlying reaction mechanisms are still rather unexplored. In particular, it is unclear how chemical interactions between the molecular reactants and the metal surface atoms are affecting reaction pathways, which makes it difficult to predict the reaction outcome. For example, it has been observed that identical molecular precursors can form different

[1]Institute of Applied Physics (IAP), Justus Liebig University Giessen, Giessen, Germany. [2]Center for Materials Research (LaMa), Justus Liebig University Giessen, Giessen, Germany. [3]Institute of Physical Chemistry, Justus Liebig University Giessen, Giessen, Germany. [4]Department of Applied Physics, Aalto University, Aalto, Finland. [5]Institute of Organic Chemistry, Justus Liebig University Giessen, Giessen, Germany. [6]Institute for Technical and Environmental Chemistry, Friedrich Schiller University Jena, Jena, Germany. [7]Helmholtz Institute for Polymers in Energy Applications Jena (HIPOLE Jena), Jena, Germany. [8]Helmholtz-Zentrum Berlin für Materialien und Energie GmbH (HZB), Berlin, Germany. ✉e-mail: doreen.mollenhauer@uni-jena.de; Daniel.Ebeling@ap.physik.uni-giessen.de

products on different surfaces depending on the surface material and its facet[20,21].

Therefore, many studies are performed that address the question of how chemical molecule-surface interactions influence the different on-surface reaction steps such as activation, diffusion, rotation, alignment, and coupling of the reactants[22–33]. Particularly relevant for such studies are coinage metal surfaces as these are commonly utilized in on-surface synthesis for creating spatially confined catalytic reaction environments. Great progress in the field of on-surface synthesis was achieved through the introduction of the so-called bond-imaging atomic force microscopy (AFM)[34] that employs carbon monoxide (CO) terminated tips to increase the lateral resolution. Bond-imaging AFM is usually performed at low temperatures under ultrahigh vacuum (UHV) conditions and has become an invaluable tool for this field of research as it enables identifying chemical structures and adsorption conformations of individual molecular precursors, intermediates, and products. Therewith, the bond-imaging technique is ideally suited for systematically studying on-surface reactions and self-assembly processes at the single-molecule level[35–46]. The method even enables the precise determination of molecular adsorption sites, which is essential for a detailed comparison with calculated molecular adsorption structures[47–51].

In this study, we focus on the spatial dependence, the strength, and the origin of the chemical interactions that an adsorbed molecule will encounter on a coinage metal surface. Useful qualitative information about the catalytic reaction environment of a metal surface is already revealed by AFM images. In the literature, Cu(111) surface atoms (top sites) usually appear as dark regions when imaged with CO tips in constant height mode, i.e., the image contrast is inverted[25,48,50,52]. Top sites of Ag(111) and Au(111) surfaces, however, appear as bright regions (non-inverted contrast)[25,49,53,54]. In case the contrast is caused by net repulsive tip-surface interactions, which are usually dominated by Pauli repulsion and short range electrostatics[55,56], a non-inverted contrast is expected. The opposite image contrasts thus reflect the higher reactivity of Cu surface atoms in comparison to Ag and Au surface atoms. Furthermore, this indicates that chemical interactions between the CO tip and the different metal surface atoms play a significant role for the imaging process. Though, a quantification and the origin of the site-specific chemical interactions remain elusive so far. In addition, it is unclear how the flexibility of the CO tip, which is known to cause image distortions and contrast inversions when imaging organic molecules[32,44,46,57–63], flat surfaces and vacancies[55,64–67], single adatoms[52,68], or adsorbed CO molecules[69], affects the image contrast. Even for metal and Si tips, which have a higher lateral stiffness than CO tips, image distortions and contrast inversions were observed[64,70–75] and it was discussed that tip relaxations caused by tip-surface forces can also be responsible for contrast inversions of scanning tunneling microscopy (STM) images[74]. So the dilemma is that although it is well known that the bending of the CO tip has a significant influence on the image contrast it is not clear how the tip behaves for a specific system (surface or molecule) and how this effect can be eliminated to obtain reliable information about the tip-sample interactions.

Here, we report systematic AFM imaging and force vs. distance measurements with CO-terminated tips on Cu(111), Ag(111), and Au(111) surfaces to reveal insights into the nature of their chemical interactions and the contrast formation process. A strong dependence of the site-specific interactions on the tip-surface distance is observed, which allows to estimate the bending of the CO tip and reduce its influence on data interpretation. With density functional theory (DFT) calculations and the subsequent use of the crystal orbital overlap population (COOP) approach, which is suitable for analyzing covalent binding contributions, we rationalize that the observed site-specific interactions correlate with covalent bonding interactions that arise from the orbital overlap of the $p$-states of the CO tip (mainly of the oxygen) and the $p$- and $d$-states of the metal surface atoms. This demonstrates that

these small covalent bonding contributions of metal surface atoms in the background of the overall dispersion-dominated interaction are accessible by the CO tip. So far, the CO tip has been regarded as a rather inert tip functionalization that can provide information about chemical interactions only in the case of highly reactive adatoms[68]. Here, we extend this to the case of far less reactive highly coordinated metal surface atoms, which is particularly interesting for on-surface chemical reactions.

Using CO tips, we measured the site-specific chemical interactions in all three special directions, which is useful for understanding on-surface reaction processes as this provides a picture for the catalytic reaction environment that a molecular reactant encounters on the surface. From the literature we already know, for example, that different chemical interactions lead to rather different activation temperatures of molecular precursors[76] or that the almost planar adsorption orientation of molecular radicals on an inert NaCl surface can be a key element for intermolecular coupling[29]. Therefore, revealing the nature of site-specific chemical interactions and their quantification is important for learning how to use different surfaces as a control knob for steering on-surface reactions and designing functional materials.

In systematic follow-up experiments the CO tip could be replaced by other tip functionalizations to probe different molecule-metal interactions. For example, using halogenated AFM tips or tips terminated with halogenated organic molecules would enable to obtain in-depth information about halogen-metal interactions. These are responsible for the catalytic dehalogenation process during on-surface Ullmann-type coupling as they weaken the halogen-carbon bond in the precursors. Or in other words, systematically determining the strength and the origin of halogen-metal interactions at different atomic sites via our approach may reveal useful trends that can help to decipher the mechanisms of dehalogenation reactions of molecular precursors. Meanwhile many different tip terminations exist (such as halogens Cl[34], Br[77], or I[38], noble gases Xe[31,61,77,78] or Kr[77], copper oxide $CuO_x$[44,62], NO[77], $CH_4$[78], NTCDI[79], H[80], or $N_2O$[81]), which are useful for such purpose due to their different reactivity, electronic configuration, charge distribution, lateral stiffness, etc. Halogens typically used in the on-surface Ullman reactions, such as Cl, Br, or I can, for example, be picked up by a metal tip relatively easily from the NaCl surface and can be used for imaging and spectroscopy measurements[29,38,77,82].

Such experiments may also provide useful information for industrial catalytic processes such as the Fischer-Tropsch process that converts a mixture of CO and $H_2$ into hydrocarbons. Catalysts containing Cu, Ag, Au, or Pt as promoters are commonly applied, for example, for the water-gas shift reaction, which controls the $CO/H_2$ ratio. The influence of site-specific interactions on the catalytic activity of different surfaces and the involved reaction mechanisms, however, remain unclear[83–87]. We would like to emphasize that revealing such information via low-temperature AFM spectroscopy measurements is not straightforward as the measured chemical interactions are highly system specific (i.e., they depend strongly on the structure of tip, surface, environment, temperature, etc.) and cannot be generalized. Nevertheless, such experiments can reveal important trends as they enable a systematic comparison of different surface materials, facets [e.g., (100), (110), etc.], and atomic sites, with different AFM-tip functionalizations. These trends are potentially useful for rationalizing possible reaction pathways and mechanisms. In addition, experimental data about site-specific chemical interactions between small molecules and metal surface atoms can serve as meaningful input parameters for theoretical studies of catalytic processes to increase their reliability.

This work also solves caveats regarding the determination of molecular adsorption sites. In ref. 40 we studied the pathway of the on-surface Ullmann reaction on Cu(111). At that time we were puzzled about the precise adsorption positions of the reaction intermediates as the identification of the atomic sites is ambiguous due to the unclear

AFM image contrast on Cu(111) (at the chosen imaging distance bright and dark images features were visible, which could due to their symmetry both be assigned to top sites). Our results presented here show that (i) the dark features that are observed in AFM frequency images of Cu(111) surface atoms with CO tips at rather large tip-surface distances correspond to Cu top sites, which was previously only an assumption and (ii) the bending of the CO as a function of the tip-surface distance is so significant that it must be taken into account for a reliable determination of molecular adsorption positions. A reliable experimental determination of molecular adsorption sites is essential for a direct comparison with computed adsorption structures. This underpins theoretical findings, such as adsorption energies, reaction barriers, charge transfers etc. which are very helpful for identifying on-surface reaction pathways[26,40,50,88]. Furthermore, the experimentally determined adsorption positions can, in turn, serve as a reliable starting point for theoretical calculations. Currently, the most favorable adsorption position (and corresponding structure) must often be identified manually or through computationally demanding global minimum search algorithms.

## Results and discussion
### Distance dependence of the image contrast on Cu(111)
Figure 1a–c shows constant-height AFM images of a Cu(111) surface for three different tip-surface distances that were measured with an oscillation amplitude of ≈ 34 pm. At the largest tip-surface separation (Fig. 1a), the Cu atoms (red circles) appear as dark regions (inverted contrast). When decreasing the tip-surface distance by 30 pm (Fig. 1b) an intermediate contrast with dark and bright features is observed.

After decreasing the distance by another 30 pm (Fig. 1c) a non-inverted contrast is observed in which the bright features are most pronounced. As indicated by the orange circles in Fig. 1b, c the dark image features are shifting in the lateral direction with decreasing tip-surface distance. The lateral shift of the atomic image features is on the order of 120 pm for Fig. 1c (see orange vs. red circles). This lateral shift that is caused by the bending of CO tip is fully reversible and not related to thermal lateral drift or piezo creep as demonstrated by Fig. 1d. The image was taken after the series of Fig. 1a–c at the same distance as in Fig. 1a and reveals a total lateral drift of about 20 pm during the whole measurement (see orange vs. red circles in Fig. 1d), which is negligible in comparison to the lateral shift caused by CO bending.

Please note that the images require relatively long sampling rates due to the low frequency shift contrast (in the sub-1 Hz regime, see color scales in Fig. 1a–c). To obtain reliable data the system was well equilibrated for this measurement (lateral drift rate ≈ 9 pm h$^{-1}$). We also repeated the measurements with different amplitudes and different CO tips and found a very similar behavior. Supplementary Fig. 1 shows a series of images that were measured with the same CO tip but for an oscillation amplitude of ≈ 137 pm. Supplementary Fig. 2 shows further images from another series that was measured with the same CO tip and amplitude as in Fig. 1, but for more distances (series of 41 images over ≈ 24 h). Supplementary Fig. 2d reveals the "first detectable image contrast" that was observed at −354 pm and is only composed of very faint dark image features (see red dots in Supplementary Fig. 2d). The bright image features (see pink dots in Supplementary Fig. 2a–c) only become visible at smaller imaging distances. For example, also in Fig. 1a (at −411 pm) already some very faint bright features are discernable (see pink circles) next to the (at this imaging distance) more pronounced dark features.

To reveal the site-specific interactions, frequency shift vs. tip-surface distance curves were measured above Cu(111) top, hollow, and bridge sites (see Fig. 1e). The curves in the inset confirm the observed image contrasts in Fig. 1a–c. At −411 pm (right dashed line in inset) the frequency shift measured at the top site (red curve) is lower than for hollow and bridge sites (orange and blue curves), which leads to the inverted image contrast in Fig. 1a. At −471 pm (left dashed line in inset), the opposite is true, which leads to the non-inverted contrast in Fig. 1c. The site-specific tip-surface force and potential energy that were calculated from the frequency shift vs. distance curves are discussed below.

### Material dependent image contrast and tip-surface forces
Next, we focus on the different image contrast observed on Cu(111), Ag(111), and Au(111) surfaces (Fig. 2a–c). Please note that the tips used are presumably Cu-CO, Ag-CO and Au-CO tips, depending on the surface material. At rather large distances, Cu(111) and Ag(111) top sites (red circles) appear as dark features in the AFM frequency shift images (inverted contrast in Fig. 2a, b). On Au(111), however, the contrast is non-inverted (Fig. 2c). Further constant-height images of Ag(111) and Au(111) at different distances are depicted in Supplementary Figs. 3 and 4. The images on Ag(111) and Au(111) are extracted from 3D grid spectroscopy data sets instead of taking a series of subsequent constant-height images as shown in Fig. 1a–d. 3D grid spectroscopy data sets are advantageous for the next part where we systematically analyze the bending of the CO tip, i.e., the dependence of the lateral shift of the atomic features on the tip-surface distance. This information can be excellently extracted from 2D slices of the 3D data (e.g., xz-slices) as this minimizes the influence of lateral sample drift and piezo creep because the data acquisition time for a single 2D slice is much shorter than for a series of images.

Figure 2d–f shows such 2D frequency shift maps that were either separately measured (in case of Cu) or extracted from 3D data sets (Ag and Au) along the directions indicated by red arrows in Fig. 2a–c. These directions were chosen in such a way that the lateral shift of the atomic

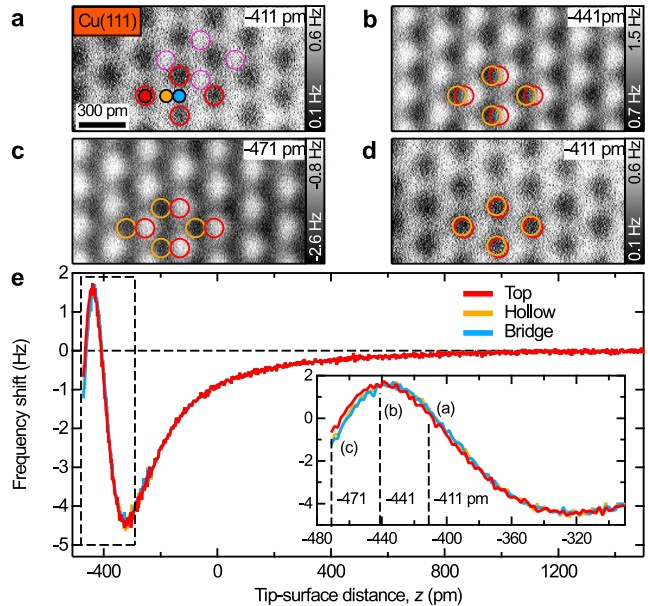

**Fig. 1 | Distance dependence of image contrast on Cu(111). a–d** Constant-height frequency shift AFM images of a Cu(111) surface measured with a CO tip. The average tip-surface distances were −411 pm (**a**), −441 pm (**b**), and −471 pm (**c**) with respect to the tunneling gap at $I_{tunnel} = 10$ pA and $U_{sample} = 100$ mV (amplitude ≈ 34 pm, resonant frequency ≈ 26960 Hz, quality factor ≈ 15791). The red circles in **a** are fitted to the dark regions and indicate the positions of Cu atoms (top sites). These dark features are the most pronounced image features at the tip-surface distance of −411 pm. The pink circles in **a** indicate very faint bright features that are already observable at the rather large tip-surface distance of −411 pm. The red circles were copied to (**b–d**). The orange circles in (**b–d**) are fitted to the dark image features and reveal the lateral shift with respect to the red circles. The image in (**d**) was measured after (**a–c**) again at −411 pm. **e** Frequency shift vs. distance curves above top (red), hollow (orange), and bridge sites (blue) (see markers in **a**). The inset in **e** shows a zoom-in (see dashed black box). The dashed black lines in the inset correspond to the distances used in (**a–c**).

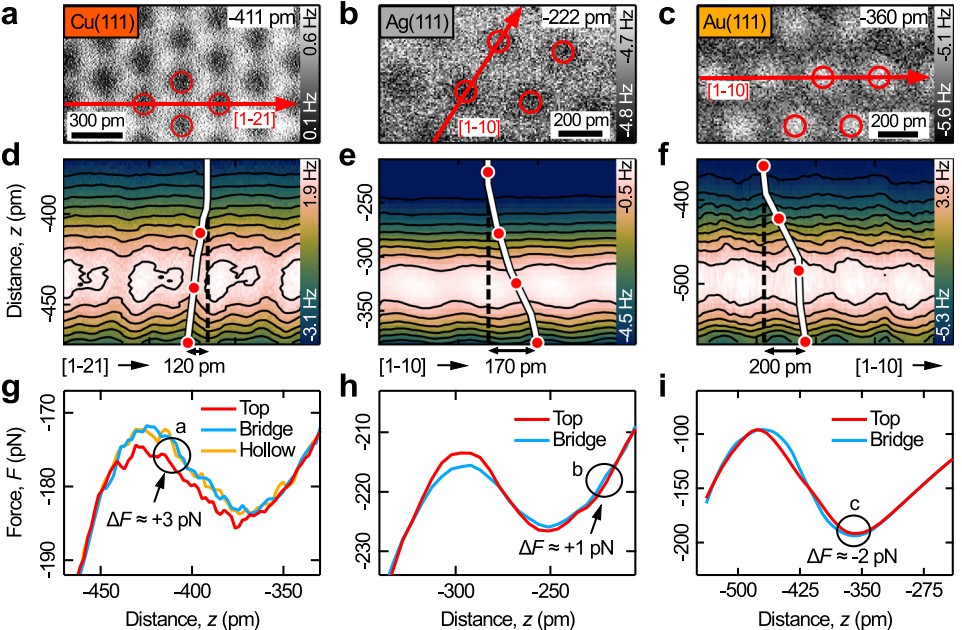

**Fig. 2 | Material dependent image contrast, CO-tip bending, and site-specific tip-surface forces. a–c** Constant-height AFM images of Cu(111), Ag(111) and Au(111) surfaces measured with CO tips. The red circles indicate top sites. The tip-surface distances are given with respect to the tunneling gap at $I_{tunnel} = 10\,pA$ (**a**, **c**), 18 pA (**b**) and $U_{sample} = 100\,mV$ (**a**, **c**), 7 mV (**b**), and amplitudes of 34 pm (**a**), 55 pm (**b**), 85 pm (**c**). **d–f** 2D frequency shift vs. distance maps measured along red arrows in (**a–c**). The directions were chosen to be almost parallel to the bending direction of the CO tip in each case. The white lines in (**d–f**) indicate the lateral movements of the atomic features (caused by bending of the CO tip) with respect to the vertical dashed black lines. The red markers in (**d–f**) indicate the tip-surface distance for the images in Figs. 1a–c, 2a–c, and Supplementary Figs. 3a–d, 4a–d. **g–i** Force vs. distance curves for top (red), bridge (blue), and hollow sites (orange) that were calculated from **d–f** via the Sader method. The "$\Delta F$ = force(bridge site) – force(top site)" values give the force difference between top and bridge sites for the distances used in (**a–c**). Other parameters: Cu (see caption Fig. 1), Ag (resonant frequency ≈ 26968 Hz, quality factor ≈ 37643), Au (resonant frequency ≈ 25860 Hz, quality factor ≈ 23368).

image features (i.e., the bending direction of the CO tip) is nearly parallel to them. The black contour lines in the 2D maps indicate neighboring points of equal frequency shift. These contours represent the atomic corrugations that would be observed in constant frequency shift scans. The white lines in Fig. 2d–f follow the atomic image features and reveal their lateral shift with decreasing tip-surface distance. The red markers on the white lines indicate the average tip-surface distances of the constant-height images in Figs. 1a–c, 2a–c, and Supplementary Figs. 3a–d, 4a–d, respectively. The lateral shifts of the atomic features are on the order of 1.3–2.2 pm per pm of vertical tip movement (corresponding to the different slopes of the white lines), which results in maximum lateral shifts (CO bending) of 120–200 pm in the presented exemplary frequency shift maps in Fig. 2d–f.

In Fig. 2g–i force vs. distance curves measured above top (red), bridge (blue), and hollow sites (orange) are depicted for the three surfaces. These curves were calculated via the Sader formula[89] from the measured frequency shift vs. distance maps. Therefore, the frequency shift maps were extended by a long-range part from a frequency shift vs. distance curve that was measured up to a tip-surface distance of ≈ 2 nm (where the frequency shift is negligible). The general trend of the curves for decreasing tip-surface distance reveals a force minimum that is followed by a force maximum. This general trend is also observed for organic molecules[49,90]. The force maximum can be rationalized by short range repulsive forces (Pauli repulsion and electrostatic interactions), which act on the flexible CO tip and tend to bend it away from the surface. In Supplementary Fig. 1k, force vs. distance curves measured with different oscillation amplitudes ranging from 34 to 187 pm are depicted, which shows that the general trend of the force curves is independent of the oscillation amplitude[91–95].

More interestingly, the force curves in Fig. 2g–i reveal also site-specific differences that are caused by different chemical interactions. For example, at the tip-surface distances marked by the black circles

(which correspond to the relatively large imaging distances used in Fig. 2a–c) the following differences between top sites (red), hollow sites (orange) and bridge sites (blue) are observed: For Cu(111) (Fig. 2g) top sites are more attractive than hollow and bridge sites. For Ag(111) top sites are more attractive than bridge sites. For Au(111), however, top sites are more repulsive than bridge sites. This agrees with the observed inverted and non-inverted frequency shift contrasts in Fig. 2a–c. The measured force differences between top sites and the other sites are on the order of $\Delta F$ = force(bridge site) – force(top site) ≈ +3 pN (Cu), +1 pN (Ag), and −2 pN (Au), respectively. These site-specific force differences are relatively small in comparison to the total attractive background caused by dispersion interactions (on the order of −200 pN, see Fig. 2g–i and Supplementary Fig. 1k). Due to the relatively small force differences the observed site-specific contrasts in the frequency shift images in Fig. 2a–c are also rather faint (see color scales, especially on Ag the contrast is on the order of only 0.1 Hz). Thus, obtaining such images requires relatively long data acquisition times.

Please note, that the absolute amounts of CO bending (lateral shifts of the image features indicated by white lines in Fig. 2d–f) and the site-specific tip-sample forces (Fig. 2g–i) also depend on the radius and shape of the metal tip, the oscillation amplitude, and other experimental conditions. However, we would like to point out that the observed qualitative behavior is rather general, as it could be reproduced with different CO tips, oscillation amplitudes and qPlus sensors. Furthermore, the finding that Cu(111) and Au(111) surfaces give inverted and non-inverted image contrasts at large distances, respectively, is also in agreement with previous frequency shift images from the literature[25,48–50,52–54].

### Site-specific force fields and potential energy landscapes
Next, we discuss the site-specific force fields and potential energy landscapes for the Cu(111) and the Au(111) surface (see Fig. 3). The total

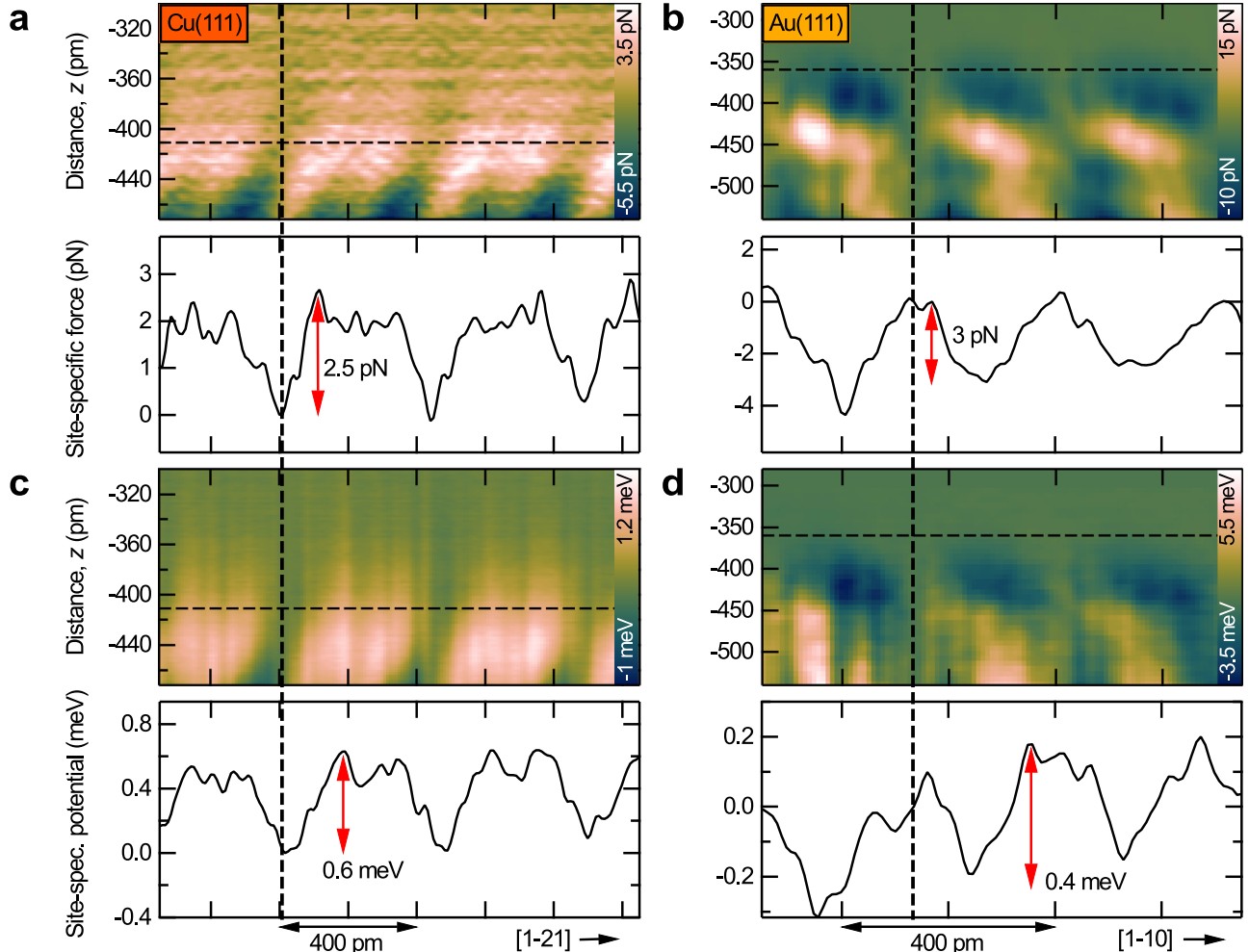

**Fig. 3 | Site-specific force fields and potential energy landscapes. a, b** 2D maps of site-specific force and individual force profiles of Cu(111) and Au(111) surfaces measured with CO tips. The 2D maps were measured along the [1-21] and the [1−10] directions of Cu(111) and Au(111), respectively (see red arrows in Fig. 2a, c). The site-specific interactions were calculated by subtracting the force measured above a top site (see black vertical dashed lines) from the 2D maps of the total interaction force (Supplementary Fig. 5a, b). The horizontal dashed lines indicate the positions of the individual force profiles shown below the 2D maps. The tip-surface distances for the profiles are the same as in Fig. 2a, c (−411 pm for Cu and −360 pm for Au). **c, d** Corresponding 2D site-specific potential energy landscapes and individual potential energy profiles.

force and potential energy maps that were calculated from the frequency shift maps in Fig. 2d, f are shown in Supplementary Fig. 5. To reveal the site-specific interactions, we subtracted the force and potential energy vs. distance curves measured above a top site (see vertical dashed lines in Fig. 3) from the corresponding 2D maps. This removes all non-site-specific components (mainly the long-range part caused by dispersion interactions) from the 2D interaction maps. The resulting 2D site-specific force and potential energy fields are shown in Fig. 3.

Below each 2D force and potential energy map in Fig. 3 individual force and potential energy profiles are depicted that were extracted along the horizontal dashed lines and correspond to the same distances as the images in Fig. 2a, c. These profiles reveal that Cu top sites are the most attractive regions with the lowest potential energy on the Cu(111) surface. The forces and potential energies measured at Cu top sites (vertical dashed line in Fig. 3a, c) and a distance of −411 pm are about 2–3 pN and 0.4–0.6 meV lower than at the other sites, respectively (see profiles in Fig. 3a, c). For the Au(111) surface the opposite is observed. Au top sites (vertical dashed line in Fig. 3b, d) are the most repulsive regions with the highest potential energy (see profiles in Fig. 3b, d). The order of magnitude here is also in the range of only a few pN and below 1 meV at the corresponding relatively large tip-surface distance of -360 pm.

## Calculated tip-surface interactions

To rationalize the origin of the measured site-specific interactions we performed DFT-D3(BJ) calculations. Figure 4a–c shows calculated force vs. distance curves for the CO tip in interaction with the three different metal surfaces. Here, the maximum at small tip-surface distances is not observed. This is due to the perfect symmetry of the modeled AFM tip, which prevents a bending of the CO away from the surface at small distances. By introducing a small asymmetry (e.g., by slightly tilting the CO-metal tip so that it is no longer perpendicular to the *ab* plane), we can reproduce the bending of the CO molecule and the shape of the experimental curves at small tip-surface distances (see Supplementary Fig. 6), whereas at larger tip-surface distances the behavior is identical to the perfectly symmetric system. Here, however, we are interested in the contribution of chemical interactions between the metal surface atoms and the CO tip at rather large distances where the bending of the CO is negligible.

In agreement with the experiments, Cu(111) top sites appear to be more attractive to the CO tip than bridge, fcc, and hcp sites (by a few pN, see Fig. 4a), while Au(111) top sites are more repulsive (Fig. 4c). In case of Ag(111) the absolute force difference between different sites is rather small, which agrees with the particularly faint contrast observed in the experiment. Though, the experiments for Ag(111) revealed a

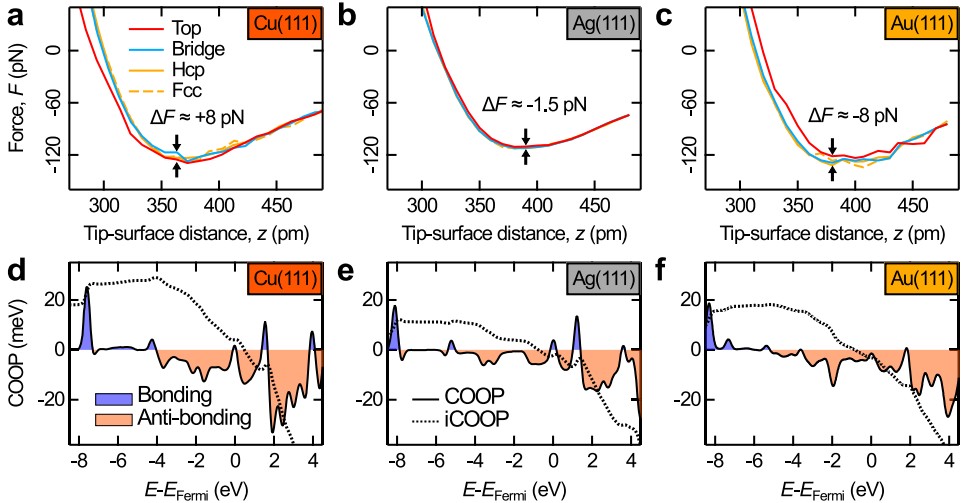

**Fig. 4 | Calculated tip-surface forces and COOP. a–c** Calculated force vs. distance curves for Cu(111), Ag(111), and Au(111) [PBE-D3(BJ)/pw (PAW)]. The curves above top, bridge, and hollow sites (fcc and hcp) are colored in red, blue, and orange, respectively. The "$\Delta F$ = force(bridge site) − force(top site)" values give the force difference between top and bridge sites. **d–f** COOP (black solid line) as a function of energy for the minimum of the potential energy vs. distance curves above the top position [tip-surface distance = 284 pm on Cu(111), 305 pm on Ag(111), and 322 pm on Au(111)]. Binding interactions (COOP > 0) are colored in blue, anti-binding interactions (COOP < 0) in orange. The black dashed lines represent the iCOOP.

positive force difference ($\Delta F$ = force(bridge site) − force(top site), see inverted contrast, Fig. 2b), while the DFT calculations give a slightly negative value. Also, using a larger Ag tip (see Supplementary Fig. 7) yields similar results, demonstrating that the CO molecule at the apex has a greater impact than size of the metal cluster of the tip[96]. This suggests that we may approach the limitations of DFT in capturing such subtle force differences.

To reveal the origin of the observed site-specific image contrasts, we performed a computational binding analysis for the small chemical interaction contributions between the CO tip and the different surfaces using the COOP method. The COOP serves as a binding indicator for modeled solids and surfaces and the integrated COOP (iCOOP) of all filled states can be interpreted analogously to the molecular bond order. This method illuminates a small covalent bonding contribution (orbital interaction) to the total interaction between the CO tip and the metal surfaces, which is dominated by dispersion forces.

Figure 4d–f depicts the projected COOP of all orbitals of the oxygen atom with the closest metal surface atom (O-M) as a function of energy as well as the respective iCOOP (dashed lines). The binding interactions (shown in blue, COOP > 0) mainly arise from overlapping of the $p$-states of the oxygen and the $p$- and $d$-states of the metal surface atoms (see Supplementary Fig. 8). Please note that the analysis of the binding described here refers to very small effects as we intentionally chose relatively large tip-surface distances to study the onset of the chemical interactions and minimize the impact of the bending of the CO tip.

Figure 5 shows a zoom-in of the iCOOP around the Fermi level. The iCOOP up to around −1 eV is positive and therefore shows that the binding contributions very slightly outweigh the anti-binding contributions for all surfaces with the largest positive value for the CO-Cu interaction. Due to predominantly anti-binding regions between −1 eV and Fermi level, the iCOOP decreases and develops into a situation where the anti-binding contributions very slightly outweigh the binding contributions for Ag(111) and Au(111). In contrast, for the Cu(111) surface the iCOOP value up to the Fermi energy is positive (binding). Furthermore, for Cu(111) and Ag(111) surfaces, a very small single peak of binding character [shaded in orange (Cu) and gray (Ag)] at the Fermi energy leads to an upwards bend of the iCOOP curves. In summary, the iCOOP value including all states up to the Fermi energy is positive for the CO-Cu(111) interaction and negative for Ag(111) followed by Au(111).

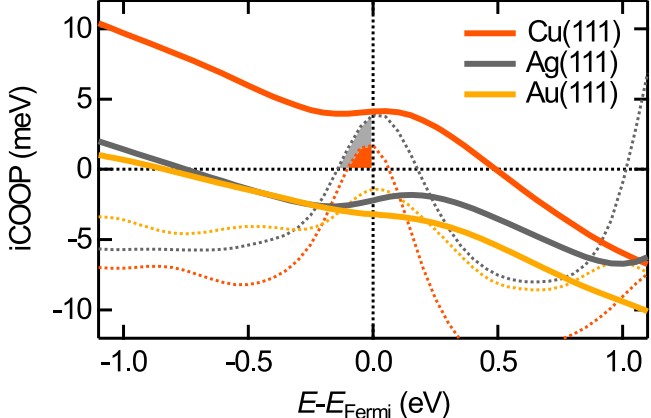

**Fig. 5 | iCOOP and COOP close to the Fermi level.** Zoom-ins of the iCOOP (solid lines) and COOP (dashed lines) for Cu(111) (orange), Ag(111) (gray), and Au(111) (yellow) as shown in Fig. 4d–f. The filled binding states close to the Fermi level are shaded in orange (Cu) and gray (Ag).

This trend agrees with the inverted image contrast for the CO-Cu(111) interaction due to the overall attractive and very small chemical O-M interaction at the top site, whereas the O-M covalent contribution for Au(111) and Ag(111) surfaces are slightly repulsive. Furthermore, this agrees with the well-known reactivity row of Cu > Ag > Au.

Recently, chemical bond formation between CO tips and Si, Cu, and Fe adatoms on Cu(111) was reported[68]. For these adatoms, however, the observed covalent binding contributions are one to two orders of magnitude stronger than for the surface atoms studied here. This can be rationalized by the lower coordination numbers (fewer binding partners) of adatoms, which leads to their higher reactivity. Along these lines, on the rather inert surface of epitaxial graphene contrast inversion is observed with highly reactive iridium-terminated metal tips but not with inert CO tips[64]. Our findings demonstrate that the onset of bond formation with small covalent binding contributions in the dispersion-dominated interaction background is observable by relatively inert CO tips even in the case of densely packed coinage metal surface atoms with a coordination number of 9, as revealed by the reproducible site-specific image contrast for the three surface

materials and the corresponding binding analysis via the COOP method.

Please note that looking into such tiny interactions at relatively large distances is relevant as it can reveal fundamental information about the origin of chemical interactions between surfaces and adsorbates. Furthermore, such findings are useful for underpinning theoretical computations, which are essential for interpreting experimental results. Since the probed tip-surface distance regime is comparable to the adsorption distance of organic molecules, the observed strength of the interactions is expected to be of the same order of magnitude. As on-surface synthesis is usually applied under UHV conditions with relatively low molecular coverages and certain reaction steps already happen at relatively low temperatures, for example, thermally activated deiodination on Cu(111) already happens below 150 K[26], the results obtained in this study are directly relevant for such processes. Operando conditions in the field of heterogeneous catalysis are, however, usually far away from UHV and low temperatures. Therefore, it will be a task for the future to extend such experimental and theoretical investigations to wider temperature and pressure ranges[97–99]. Additionally, although such small energy contributions might be negligible in describing the thermodynamics of a binding event, they can get highly relevant in kinetic arguments, e.g., for the selectivity in catalytic processes.

In summary, our measurements together with theoretical calculations offer insights into the nature, the spatial distribution and the strength of chemical interactions between CO-terminated AFM tips and metal surface atoms. It is revealed that the onset of bond formation with small covalent binding contribution is responsible for the observed inverted and non-inverted AFM image contrasts, the site-specific force fields and the potential energy landscapes at relatively large distances. The ability to systematically analyze such small chemical interactions between CO tips and metal surface atoms in the background of the overall dispersion dominated interaction is important for understanding on-surface reaction mechanisms, such as diffusion, activation and coupling processes as this gives a picture for the force and potential energy landscapes that molecular reactants encounter after adsorption on a metal surface. In the future this concept could be extended by using tailored-made tip terminations for studying further chemical interactions. For example, halogen terminated tips could be used to study halogen-metal interactions, which are responsible for the dehalogenation process that is part of the on-surface Ullmann type coupling. Such experiments can be systematically performed with different tip terminations for different surface materials, facets and atomic sites, which allows their differences to be analyzed and classified. This allows to reveal trends that will potentially help decipher the reaction mechanisms on-surfaces. This knowledge will be useful for steering the outcome of on-surface reactions via different surfaces and for designing organic nanomaterials.

Moreover, our findings enable a precise and reliable determination of molecular adsorption sites as we can now (i) unambiguously relate the observed image features (dark or bright regions in AFM frequency-shift images) to the atomic sites and (ii) account for the bending of the CO tip. The former assignment was previously only made on the basis of symmetry considerations[48–52] and is now being extended by evaluating the site-specific tip-surface interactions. The latter leads to the observed lateral shifting of the atomic image features as a function of the tip-surface distance. Our 2D frequency shift maps precisely reveal the distances where the CO tip starts bending as a function of tip-surface distance (see white lines in Fig. 2d–f). This shows that rather large imaging distances (i.e., distances larger than the points where the white lines in Fig. 2d–f start bending in lateral direction) should be chosen for locating surface atoms to avoid significant errors due to the bending of the CO tip. It is very important to take this effect into account as the image features of adsorbed molecules do not systematically shift in the same lateral direction as the features of the surface atoms (see Supplementary Fig. 9)[49,57]. This means that imaging distances that are too small (i.e., with significant bending of the CO tip), both on the surface and the adsorbed molecule, will not cancel each other out. Such precisely determined adsorption positions of organic molecules can also serve as a starting point for computational studies and underpin the theoretical findings, such as adsorption energies, reaction barriers, charge transfers, etc., which will potentially enable identifying on-surface reaction pathways.

## Methods
### AFM measurements
The experiments were performed with a low temperature AFM (ScientaOmicron) that utilizes qPlus sensors[100] with tungsten tips (base pressure < $10^{-10}$ mbar, temperature = 5.2 K). The different metal surfaces (Mateck) were cleaned by cycles of argon sputtering and annealing. Before functionalizing the AFM tips with CO, the metal tips were conditioned on each surface via indentations and voltage pulses. Therefore, our tips are likely covered with metal atoms of the respective surface material (Cu, Ag, or Au). This ensures a high comparability with measurements from literature as the tips are assumed to be covered with surface material after working for some time on a respective surface. We used oscillation amplitudes in the range of ≈ 30–180 pm, which is typical for bond-imaging studies and provides the highest image contrast[90]. The amplitudes were calibrated using the constant current method[92]. The reliability of the amplitude calibration was confirmed by measuring spectroscopy curves at different amplitudes (see Supplementary Fig. 1j, k), which reveals that the resulting force vs. distance curves are independent of the used amplitude calibration[91–95]. The resonant frequencies and quality factors of the qPlus sensors were on the order of 26–27 kHz and 15,000–40,000, respectively. CO was picked up from Cu(111) using standard procedures[101]. On Ag(111) and Au(111) surfaces, the tips were functionalized by imaging in STM mode or by voltage pulses.

### DFT calculations
Periodic unrestricted DFT calculations of the CO tip in interaction with the Cu(111)/Ag(111)/Au(111) surfaces have been performed at the PBE-D3(BJ)/pw (PAW) level of theory using the Vienna ab initio simulation package (version 5.4.1)[102–105]. Subsequent calculations of the projected COOP to analyze the small covalent binding contribution between the oxygen from the CO tip and the surface atom below the tip have been carried out using the LOBSTER program (version 4.0.0)[106–109]. Further computational details are given in the Supplementary Note 6.

## Data availability
The data that support the findings of this study are available from the corresponding authors upon request.

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

## Acknowledgements

Financial support by the Deutsche Forschungsgemeinschaft via the grants EB535/1–1(363901684) (D.E.), EB535/2–1 (D.E.), EB535/4-1(448547917) (D.E.), SCHI619/13(417197256) (A.Sch.), MO2995/2-1(417197256) (D.M.), and the GRK (Research Training Group) 2204 "Substitute Materials for Sustainable Energy Technologies" (D. M., A.Sch.) is gratefully acknowledged. Further financial support was provided by the LOEWE Program of Excellence of the Federal State of Hesse (LOEWE Focus Group PriOSS 'Principles of On-Surface Synthesis') (D.M., H.A.W., A.Sch.). The authors acknowledge computational resources provided by the HPC Core Facility and the HRZ of the Justus-Liebig-University Giessen (P.H., K.L.H.P., D.M.).

## Author contributions

D.E. conceived the project. J.T., Q.Z., A.Se., and D.E. performed the measurements. J.T. and D.E. analyzed the data. P.H., J.J., K.L.H.P., and D.M. performed the DFT computations. J.T., P.H., H.A.W., D.M., A.Sch. and D.E. contributed significantly to the scientific discussion of the results. All authors contributed to the manuscript.

## Funding

## Competing interests

The authors declare no competing interests.
