## [Transparent Peer Review file · Nature Communications]

Probing weak chemical interactions of metal surface atoms with CO-terminated AFM tips identifies molecular adsorption sites

Corresponding Author: Dr Daniel Ebeling

Version 0:

Reviewer comments:

Reviewer #1

(Remarks to the Author)

The article systematically studies atomic contrast in atomic force microscopy (AFM) and force spectroscopy on flat Cu, Ag, and Au surfaces, comparing experimental results with density functional theory (DFT) calculations and achieving good agreement. The paper is clearly written, easy to follow, and presents an appropriate selection of both experimental and theoretical methods. The results are reasonable and well-supported by evidence.

The primary consideration regarding the impact of the article is its relevance to a broader scientific audience. While the topic of image contrast on flat coinage metal substrates and contrast inversion has been an ongoing subject of scientific discussion in the AFM community, this study successfully places the discussion on solid theoretical and experimental grounds. The explanation for image contrast and contrast inversion—attributed to a weak coordination bond between the electron pair on CO and the top-positioned Cu atom—is logical and well-supported. Additionally, the article provides a valuable and clear discussion of CO bending and the inversion of force curves, emphasizing the importance of distinguishing these effects, as they can be easily conflated in less systematic studies. I appreciate the authors' meticulous and systematic effort in deciphering delicate chemical contrast on inert, flat coinage metal surfaces.

However, I am uncertain about the study's broader relevance to on-surface reactions and catalysis, as suggested in the introduction. The key challenge with chemical interactions between coinage metals and molecules is their strong dependence on the local chemical environment, including the type of metal, coordination number, and molecular species. Consequently, conclusions drawn from one specific system may not be easily generalizable to others, even if they are similar. The authors themselves acknowledge this limitation by demonstrating how contrast on Ag differs from that on Cu due to minute variations in the delicate balance between bonding and antibonding contributions in the crystal orbital overlap population (COOP) analysis. They also highlight the distinction between flat surfaces and low-coordinated adatoms. Given these factors, I am uncertain whether the conclusions derived from a CO-decorated tip can be extended to other molecular ligands or to catalytic reactions, such as Ullmann coupling.

Alongside the authors, I believe that additional experimental investigations—such as comparative studies using different ligands (e.g., halogen-decorated tips) or a systematic comparison of flat surfaces with gradually decreasing coordination environments (e.g., vacancies, step edges, adatoms)—would be particularly valuable for further elucidating these interactions.

In attempting to distill a clear take-home message from this study, I find that the main conclusion is the observation that chemical forces are highly system-specific and cannot be easily described by simple rules. While this is an important insight, it is also somewhat expected and, in a broader sense, not entirely satisfying.

Overall, the study provides valuable and rigorously obtained data on atomic-scale chemical interactions but leaves open questions regarding its general applicability to surface chemistry and catalysis.

Reviewer #2

(Remarks to the Author)

The paper describes the interaction of CO-molecules attached to a probing tip and the closed-packed metallic surfaces of Cu(111), Ag(111) and Au(111). The force microscopy part shows that the imaging of Cu(111) gives an inverted contrast with more attractive interactions at the top sites, whereas the imaging on Au(111) shows a non-inverted contrast on the top sites, which corresponds to repulsive interaction. In case of Ag(111), a very faint inverted contrast is found. The differences in forces are very small (some pN), but seem to be reproducible thanks to the careful work with different tips. The effect of bending of the CO molecule seems excluded. Comparison with DFT calculations show that Cu(111) has more overlap of bonding orbital compared to Au(111), where anti-bonding orbital overlap seems to dominate. This is also visualized by the crystal orbital overlap population (COOP) method and integrated COOP (iCOOP). Because of the relevance of coinage metals for catalytic reactions, specially for on-surface chemistry, this work is important. I propose to publish with minor revision.

Points to be addressed:

The authors mention that it might be of interest to investigate halogen terminated tips. Are there some results in the literature or by the authors?

It might be also interesting to mention some historic STM work on the coinage metals. Inverted contrasts were reported, possibly related to some force mediated contrast.

A possible extension of this work might be to study the other crystallographic faces (100), (110) and so on. A comment on this point, might be of interest to readers.

Reviewer #3

(Remarks to the Author)

In this work, non-contact frequency-modulated scanning force micro-spectroscopy is employed for a detailed revisit of the forces between a CO-functionalized tip and the atoms on a Cu(111), Ag(111) and Au(111) surface. For Cu(111), a clear contrast inversion is reported for Cu top vs. valley sites when comparing the frequency-shift maps [frequency shift is proportional to $-(dF/dz)$] - and by integration over the height z thus resulting in force maps $F(z, x, y)$: the Cu-top site is most repulsive of all sites at small distances (in the Pauli repulsive regime, small z) and most attractive of all sites in the attractive z -region (larger z , attractive dispersion forces + small covalent contribution). This contrast inversion is – interesting enough – absent at the Au(111) surface.

Next, the force(x, y) map is studied in more detail (at certain height z -values) by mapping over line scans. The authors claim that they can extract a small covalent contribution from the much larger overall attractive force (mostly due to dispersive forces). This small covalent part is also calculated by DFT simulations, and appears to be slightly attractive for the Cu-site, while it is (at the same Z) already repulsive in the case of the Au top atom at Au(111).

EVALUATION: This is a very detailed and well written study using state-of-the-art frequency modulated force microscopy. A part of the work is in fact a repeated investigation (but with improved accuracy), while the second part of the work, dealing with the extraction of the very small covalent M...OC-tip contribution, can be considered as a new step forward. The impact of the work lies in (1) finding the limits of state-of-the-art force microscopy and the relation to on-surface chemistry of important organic complexes with electronic, spintronic or catalytic function. I recommend publication of this work in Nature Communications.

I have minor comments that the authors could take into account in a revision:

COM1 Title: ultra-low chemical interactions of metal surface atoms...low is possibly not the best word here; my suggestions: small or tiny?

COM2 The contrast inversion seen on Cu(111) is very similar to what has been seen for C-sites on graphene on Ir(111), with similar explanation, see Ref. 61. Perhaps worth to mention this?

COM3 It could be mentioned in more detail how the metal-Cu tip is modelled for the DFT simulations. While the greater part of the force can be contributed to the CO end of the tip, the tip "metal-cluster" contributes also for a small part, see Quantitative Atomic Force Microscopy with Carbon Monoxide Terminated Tips, Phys. Rev. Lett. 106, 046104 (2011).

COM4: The authors relate the impact of their work to on-surface synthesis of molecules and catalysis. However, the potential energy differences observed between different sites (say top and valley sites on Cu(111)) are very tiny, urging the question if they are relevant in the temperature region of on-surface synthesis and catalysis. Perhaps, the authors could motivate this last part of the text better?

Reviewer #4

(Remarks to the Author)

Tschakert et al. used low-temperature AFM with a CO-tip to investigate the bare (111) surfaces of Cu, Ag and Au. They propose that they can align the AFM images with the underlying lattice. This would be very helpful, as they describe, for any studies of molecular adsorbates on these surfaces. Moreover, with the relatively inert CO tip they effectively probe the potential energy landscape of the surface and show just how flat these densely-packed surfaces are (lateral diffusion barriers less than 1 meV). Given the wide use of these surfaces, I believe such results are of interest generally in the chemistry and surface science community.

My fundamental problem with the manuscript in its current state is that I do not understand the assignment of the AFM images to the underlying lattice. For example, we are presented with an AFM image in Figure 1a and told that the dark circles correlate with atomic positions. Why? There are also lighter protrusions that, from symmetry arguments, could equivalently be the atomic positions. I expected this to be clearly explained when the measurements of potential energy as a function of height (Fig. 3) were discussed: Perhaps the atomic positions themselves would be seen via Pauli repulsion as maxima in the energy positions. But the dashed lines are over an energy minimum at the lowest height. Finally I thought that the DFT would be used to explain this. But the DFT calculations show a different assertion than the authors assert for Ag. Therefore can we believe it for Cu and Au?

My impression is that the authors have good reasons for asserting the contrast as shown. Perhaps they leaked in CO, which is known to bond to top sites on all surfaces discussed and were able to orient their images with these marker molecules. They also (Ref. 40) have at least thought about this before but assert that it is still an open question.

The discussion of the DFT results are very important and I would ask the authors to lengthen this section. Currently, they state, "the experiments for Ag(111) revealed a negative force difference [inverted contrast, Fig. 2(b)], while the DFT calculations give a slightly positive value. We rationalize that this is caused by the larger and more complex metal tip that is used in the experiments and/or the limits for DFT-D3 for describing such small force differences.". Two points: (1) How would a larger and more complex metal tip change the contrast? Are you arguing electrostatic interaction via the large metal dipole of the tip apex or that the structure of the metal tip changes the chemical binding of the CO at the apex? (2) What are the limits of DFT-D3? 2 pN? A better description would be useful for future investigations wanting to use it.

I would also note that the application to adsorbates in general requires that the adsorbate is also relatively inert and that the surface only acts as a potential energy landscape. For many catalytic molecules this is not the case, as surface interaction is what weakens bonds, etc.

I have some smaller comments that I recommend the authors consider when they have adequately modified the manuscript to address the major issue described above:

- Consider "ultra-weak" instead of "ultra-low", as low is sometimes used for a spatial coordinate
- (p. 2) The authors talk about image contrast inversion. However, this was not properly explained. vdW attraction would nominally make one expect attraction. To understand this assumption, one has to understand that your hypothesis is interaction dominated by Pauli repulsion, which is not stated.
- To make the work attractive to a broader community (chemistry), I would recommend discussing the well-known reactivity row of $\text{Cu} > \text{Ag} > \text{Au}$.
- Taking CO bending into account (p. 2) has been done before by e.g. Gross et al. Science 2012; Scheuerer et al PRL 2019; Hofmann et al New J Phys 2022
- I would ask that the authors incorporate a comment about their work in contrast to that published by Neel and Kröger, Nano Letters 2021, especially regarding the COOP analysis
- (p. 3) The authors claim "a non-inverted contrast is observed" - it could be that contrast does not change and just shifts, as indicated by the yellow circles. This relates strongly to my main critique, above
- Although I expect the answer to be no, I am curious as to whether any additional excitation was observed during measurements
- Define x and z. This will also help the figures. Currently writing "distance" on both the x and y axes (e.g. Fig 2 d-f) should read "x" or "x distance"
- Vertical dashed lines were not clearly defined in Fig. 2
- It's great that the authors checked their force-distance curves with several amplitudes. Because the curve has several inflection points, this is a nice check for ill-fitting as described in Sader et al Nat Nano 2018 - but this citation is missing.
- (p. 6) Hollow is not shown for Au or Ag in Fig. 2, so the following sentences are misleading: "Cu(111) and Ag(111) top sites (red) appear to be more attractive than hollow sites (orange) or bridge sites (blue). For Au(111), however, top sites are more repulsive than the other sites."
- Is CO bending ignored in the discussion of Figs. 3 and 4? Is that valid?
- The authors claim "By introducing a small asymmetry, we can reproduce the bending of the CO molecule and the shape of the experimental curves". This is very important and I would ask them to include a figure and discussion of this in the main text or supporting material
- (p. 9) The energy window of -8 to 4 eV seems arbitrary - can you justify it?

Version 1:

Reviewer comments:

Reviewer #1

(Remarks to the Author)

I would like to thank the authors for their thoughtful and detailed responses to my earlier comments. I maintain my view that the manuscript presents high-quality experimental work, which is carefully analyzed and methodologically sound. The force spectroscopy measurements and their comparison to DFT are of high value to the AFM community, and I support publication of the article on that basis.

That said, I remain skeptical about the broader relevance to catalysis and on-surface chemistry as currently framed in the revised manuscript. While I appreciate the authors' effort to contextualize their work by discussing connections to catalytic processes such as the Fischer-Tropsch reaction or Ullmann-type coupling, I do not find these arguments chemically well-founded.

In particular, the interactions studied here involve ultra-weak forces between a passivated CO-tip and flat coinage metal surfaces, mediated via the lone pair on the oxygen atom of the CO molecule. This is a fundamentally different regime from that relevant to catalysis, where free CO molecules typically chemisorb through the carbon atom—not oxygen—onto highly reactive sites such as adatoms, step-edges, or vacancies. These sites are generally far more catalytically active than the flat, low-index terraces investigated here. Thus, the bonding mechanisms, energy scales (meV vs eV), and relevant surface configurations are qualitatively different, making it problematic to generalize insights across these systems.

Similarly, the extrapolation to dehalogenation reactions in Ullmann-type couplings is, in my view, beyond the realistic interpretative scope of this work. Activation of such halogenated precursors is not limited to interaction with the halogen atom alone, but often involves electron donation into anti-bonding orbitals of the extended π -system of the molecule. This delocalized activation process is hard to reconcile with the localized, site-specific interactions measured in the present study. Moreover, the reactivity of halogens (Cl, Br, I) varies substantially, and their behavior on metal surfaces cannot be meaningfully deduced from force differences of a few meV observed under these highly controlled AFM conditions. Even the relative trends between coinage metals observed in the manuscript are not consistent across systems, further limiting generalizability.

The authors themselves acknowledge that such chemical forces are "highly system-specific and cannot be easily described by simple rules," a point I fully agree with. But this also implies that any attempt to draw broader conclusions about catalytic reactivity or surface chemistry mechanisms must be approached with great caution. As it stands, I feel the added sections in the revised manuscript overreach and risk compromising the otherwise rigorous and focused scientific character of the study.

While I recognize the potential of AFM methods to contribute to surface chemistry—particularly by enabling site-resolved measurements under controlled conditions—I would strongly encourage the authors to remove or significantly temper the references to catalysis and on-surface synthesis. The value of this work lies in its precise, quantitative probing of weak interactions on well-defined surfaces, and it would benefit from a sharper focus on that strength.

In summary, I support publication of the article for its core contributions to AFM spectroscopy, provided the speculative and insufficiently supported catalytic claims are corrected or removed, in order to preserve the scientific rigor and clarity of the work. Nevertheless, it is ultimately up to the editor to judge whether the impact and scope of the article are, in such case, sufficiently broad to merit publication in Nature Communications.

Reviewer #2

(Remarks to the Author)

The revised manuscript is improved and the questions of the the referees were addressed very well. I support publication in its present form.

Reviewer #3

(Remarks to the Author)

Revisions: The authors have responded to the many detailed comments of this reviewer and the others adequately, and have revised the ms. adequately with rephrasing and some additions.

Importance for a broad audience: This ms. obtained four critical and detailed review reports. Overall the tone of reviewers was supportive; the detailed and relevant comments and questions also showed the interest that this ms. will receive from the inner-circle scanning probe community. Eventually, people from the catalytic community may start to find this type of work valuable (although my personal experience is that they generally neglect this type of fine-tuned scanning probe microscopy and/or remain working with commercial apparatus with high-amplitude tip vibrations accompanied by vague "even not wrong" interpretations).

I strongly support this work to be published in Nature Communications.

Reviewer #4

(Remarks to the Author)

As reviewer 3 nicely summarized, "A part of the work is in fact a repeated investigation (but with improved accuracy), while the second part of the work, dealing with the extraction of the very small covalent M...OC-tip contribution, can be considered as a new step forward."

I have only minor comments, which can be found in the attached pdf. In short:

- 1) Recommend a citation
- 2) A comment about DFT energy accuracy

The manuscript looks very much improved and I recommend it for publication after the authors consider these minor comments.

Decision on Nature Communications manuscript NCOMMS-25-09336

Author reply

Reviewer #1 (Remarks to the Author):

The article systematically studies atomic contrast in atomic force microscopy (AFM) and force spectroscopy on flat Cu, Ag, and Au surfaces, comparing experimental results with density functional theory (DFT) calculations and achieving good agreement. The paper is clearly written, easy to follow, and presents an appropriate selection of both experimental and theoretical methods. The results are reasonable and well-supported by evidence.

The primary consideration regarding the impact of the article is its relevance to a broader scientific audience. While the topic of image contrast on flat coinage metal substrates and contrast inversion has been an ongoing subject of scientific discussion in the AFM community, this study successfully places the discussion on solid theoretical and experimental grounds. The explanation for image contrast and contrast inversion—attributed to a weak coordination bond between the electron pair on CO and the top-positioned Cu atom—is logical and well-supported. Additionally, the article provides a valuable and clear discussion of CO bending and the inversion of force curves, emphasizing the importance of distinguishing these effects, as they can be easily conflated in less systematic studies. I appreciate the authors' meticulous and systematic effort in deciphering delicate chemical contrast on inert, flat coinage metal surfaces.

However, I am uncertain about the study's broader relevance to on-surface reactions and catalysis, as suggested in the introduction. The key challenge with chemical interactions between coinage metals and molecules is their strong dependence on the local chemical environment, including the type of metal, coordination number, and molecular species. Consequently, conclusions drawn from one specific system may not be easily generalizable to others, even if they are similar. The authors themselves acknowledge this limitation by demonstrating how contrast on Ag differs from that on Cu due to minute variations in the delicate balance between bonding and antibonding contributions in the crystal orbital overlap population (COOP) analysis. They also highlight the distinction between flat surfaces and low-coordinated adatoms. Given these factors, I am uncertain whether the conclusions derived from a CO-decorated tip can be extended to other molecular ligands or to catalytic reactions, such as Ullmann coupling.

Alongside the authors, I believe that additional experimental investigations—such as comparative studies using different ligands (e.g., halogen-decorated tips) or a systematic comparison of flat surfaces with gradually decreasing coordination environments (e.g., vacancies, step edges, adatoms)—would be particularly valuable for further elucidating these interactions.

In attempting to distill a clear take-home message from this study, I find that the main conclusion is the observation that chemical forces are highly system-specific and cannot be easily described by simple rules. While this is an important insight, it is also somewhat expected and, in a broader sense, not entirely satisfying.

Overall, the study provides valuable and rigorously obtained data on atomic-scale chemical interactions but leaves open questions regarding its general applicability to surface chemistry and catalysis.

Author reply: We appreciate the valuable assessment of our manuscript. To clarify its take-home message and discuss its general applicability and importance for surface chemistry and catalysis

in more detail we made the following changes to the manuscript (see changes in the abstract, introduction page 2-4 and discussion page 11-12):

- 1) We specify the significance of site specific interactions for catalytic applications such as the Fischer-Tropsch process and the water-gas shift reaction.
- 2) We clearly point out - as suggested by the reviewer - that such chemical forces are highly system-specific and cannot be easily described by simple rules.
- 3) We emphasize the point that such AFM experiments are still very useful as they enable systematic comparison of different surface materials, facets, atomic sites, etc. with the same tip. Therefore, this study provides a general method to elucidate these parameters. This information is also useful as input parameters for theoretical investigations.
- 4) We emphasize that the unambiguous determination of molecular adsorption sites, is essential for understanding on-surface reactions as it allows a direct comparison with calculated adsorption structures (and also serves as a starting point for computations). With the comparison of Cu, Ag and Au the most important metals used in on surface reactions are investigated.

Reviewer #2 (Remarks to the Author):

The paper describes the interaction of CO-molecules attached to a probing tip and the closed-packed metallic surfaces of Cu(111), Ag(111) and Au(111). The force microscopy part shows that the imaging of Cu(111) gives an inverted contrast with more attractive interactions at the top sites, whereas the imaging on Au(111) shows a non-inverted contrast on the top sites, which corresponds to repulsive interaction. In case of Ag(111), a very faint inverted contrast is found. The differences in forces are very small (some pN), but seem to be reproducible thanks to the careful work with different tips. The effect of bending of the CO molecule seems excluded. Comparison with DFT calculations show that Cu(111) has more overlap of bonding orbital compared to Au(111), where anti-bonding orbital overlap seems to dominate. This is also visualized by the crystal orbital overlap population (COOP) method and integrated COOP (iCOOP). Because of the relevance of coinage metals for catalytic reactions, specially for on-surface chemistry, this work is important. I propose to publish with minor revision.

Points to be addressed:

The authors mention that it might be of interest to investigate halogen terminated tips. Are there - some results in the literature or by the authors?

Author reply: We appreciate the valuable assessment of our manuscript. See new sentence on page 3: " Halogens typically used in the on-surface Ullman reaction, such as Cl, Br, or I can, for example, be picked up by a metal tip relatively easily from the NaCl surface and can be used for imaging or spectroscopy measurements. 29, 38, 75, 80"

It might be also interesting to mention some historic STM work on the coinage metals. Inverted contrasts were reported, possibly related to some force mediated contrast.

Author reply: We added a citation to Ref 74 on page 3 and changed the following sentence: "Even for metal and Si tips, which have a higher lateral stiffness than CO tips, image distortions and contrast inversions were observed 64,70-75 and it was discussed that tip relaxations caused by

tip-surface forces can also be responsible for contrast inversions of scanning tunneling microscopy (STM) images. 74”

A possible extension of this work might be to study the other crystallographic faces (100), (110) and so on. A comment on this point, might be of interest to readers.

Author reply: We added a comment on page 4: “Nevertheless, such experiments enable a systematic comparison of different surface materials, facets [e.g. (100), (110), etc.], and atomic sites, with the same AFM tip, which can reveal important differences and trends that are useful for identifying possible reaction mechanisms.”

And on page 12: “Such experiments can be systematically performed with the same tip termination for different surface materials, facets and atomic sites, which allows their differences to be analyzed and classified”

Reviewer #3 (Remarks to the Author):

In this work, non-contact frequency-modulated scanning force micro-spectroscopy is employed for a detailed revisit of the forces between a CO-functionalized tip and the atoms on a Cu(111), Ag(111) and Au(111) surface. For Cu(111), a clear contrast inversion is reported for Cu top vs. valley sites when comparing the frequency-shift maps [frequency shift is proportional to $-(dF/dz)$] - and by integration over the height z thus resulting in force maps $F(z, x, y)$: the Cu-top site is most repulsive of all sites at small distances (in the Pauli repulsive regime, small z) and most attractive of all sites in the attractive z -region (larger z , attractive dispersion forces + small covalent contribution). This contrast inversion is – interesting enough – absent at the Au(111) surface.

Next, the force(x, y) map is studied in more detail (at certain height z -values) by mapping over line scans. The authors claim that they can extract a small covalent contribution from the much larger overall attractive force (mostly due to dispersive forces). This small covalent part is also calculated by DFT simulations, and appears to be slightly attractive for the Cu-site, while it is (at the same Z) already repulsive in the case of the Au top atom at Au(111).

EVALUATION: This is a very detailed and well written study using state-of-the-art frequency modulated force microscopy. A part of the work is in fact a repeated investigation (but with improved accuracy), while the second part of the work, dealing with the extraction of the very small covalent M...OC-tip contribution, can be considered as a new step forward. The impact of the work lies in (1) finding the limits of state-of-the-art force microscopy and the relation to on-surface chemistry of important organic complexes with electronic, spintronic or catalytic function. I recommend publication of this work in Nature Communications.

I have minor comments that the authors could take into account in a revision:

COM1 Title: ultra-low chemical interactions of metal surface atoms...low is possibly not the best word here; my suggestions: small or tiny?

Author reply: We appreciate the valuable assessment of our manuscript. We replaced “ultra-low” by “ultra-weak” and “ultra-small” throughout the text.

COM2 The contrast inversion seen on Cu(111) is very similar to what has been seen for C-sites on graphene on Ir(111), with similar explanation, see Ref. 61. Perhaps worth to mention this?

Author reply: We added the following sentence on page 11: Along these lines, on the rather inert surface of epitaxial graphene contrast inversion was observed in case of highly reactive iridium-terminated metal tips but not with inert CO tips. 64 (was Ref 61 before)

COM3 It could be mentioned in more detail how the metal-Cu tip is modelled for the DFT simulations. While the greater part of the force can be contributed to the CO end of the tip, the tip “metal-cluster” contributes also for a small part, see Quantitative Atomic Force Microscopy with Carbon Monoxide Terminated Tips, Phys. Rev. Lett. 106, 046104 (2011)

Author reply: We greatly appreciate the reviewers’ valuable feedback and sincerely apologize for any lack of clarity in the original manuscript. A detailed description of the DFT methodology, as well as the structures of the Cu(111), Ag(111), and Au(111) surfaces and the corresponding metal–CO tips used, is provided in the Supporting Information under the section entitled “Modelling of the CO tip–metal surface system.” In the revised manuscript, we have now included a clear reference to this section.

In response to Reviewer #4’s comment, we also performed additional calculations using larger tips comprising three layers (consisting of one, three, and six atoms, with the six atoms fixed to represent the bulk). These simulations yielded results consistent with those obtained using smaller tips, differing only by a small, approximately constant shift in force. Furthermore, our findings confirm that the interaction with the surface is dominated by the CO molecule, in agreement with the study recommended by the reviewer (Sun et al. <https://doi.org/10.1103/PhysRevLett.106.046104>).

COM4: The authors relate the impact of their work to on-surface synthesis of molecules and catalysis. However, the potential energy differences observed between different sites (say top and valley sites on Cu(111)) are very tiny, urging the question if they are relevant in the temperature region of on-surface synthesis and catalysis. Perhaps, the authors could motivate this last part of the text better?

Author reply: We added a paragraph to the discussion (see page 11): “Please note that looking into such ultra-weak interactions at relatively large distances is relevant as it can reveal fundamental information about chemical interactions between surfaces and adsorbates. Furthermore, such findings are useful for underpinning theoretical computations, which is essential for interpreting the experimental results. Since the probed tip-surface distance regime is comparable to the adsorption distance of organic molecules, the observed strength of the interactions is expected to be of the same order of magnitude. As on-surface synthesis is usually applied under UHV conditions with relatively low molecular coverages and certain reaction steps already happen at relatively low temperatures, for example, thermally activated deiodination on Cu(111) already happens below 150 K, 26 the results obtained in this study are directly relevant for such processes. As operando conditions in the field of heterogeneous catalysis far away from UHV and low temperatures it will be a task for the future to extend such experimental and theoretical investigations to wider temperature and pressure ranges. 96-98 Additionally, although such small energy contributions might be negligible in describing the thermodynamics of a binding event, they can get highly relevant in kinetic arguments, e.g. selectivity in catalytic processes.”

Reviewer #4 (Remarks to the Author):

Tschakert et al. used low-temperature AFM with a CO-tip to investigate the bare (111) surfaces of Cu, Ag and Au. They propose that they can align the AFM images with the underlying lattice. This would be very helpful, as they describe, for any studies of molecular adsorbates on these surfaces. Moreover, with the relatively inert CO tip they effectively probe the potential energy landscape of the surface and show just how flat these densely-packed surfaces are (lateral diffusion barriers less than 1 meV). Given the wide use of these surfaces, I believe such results are of interest generally in the chemistry and surface science community.

My fundamental problem with the manuscript in its current state is that I do not understand the assignment of the AFM images to the underlying lattice. For example, we are presented with an AFM image in Figure 1a and told that the dark circles correlate with atomic positions. Why? There are also lighter protrusions that, from symmetry arguments, could equivalently be the atomic positions. I expected this to be clearly explained when the measurements of potential energy as a function of height (Fig. 3) were discussed: Perhaps the atomic positions themselves would be seen via Pauli repulsion as maxima in the energy positions. But the dashed lines are over an energy minimum at the lowest height. Finally I thought that the DFT would be used to explain this. But the DFT calculations show a different assertion than the authors assert for Ag. Therefore can we believe it for Cu and Au?

Author reply: We appreciate the valuable assessment of our manuscript. We also have another series of images of Cu(111) with the same CO-tip, which contains 41 images at different distances and took about 24 h. The images shown in the manuscript were measured in a separate series to ensure that the effect of thermal drift is as low as possible. In the supporting information we added images and spectroscopy curves from the other series. Please see Fig. S2d, which was taken at $z = -354$ pm (about 60 pm more far away than Fig. 1a in the manuscript). The first visible contrast that appears at rather large distances is only composed of (very faint) dark features. Hitherto the dark features were assigned to top sites only because of symmetry reasons. Our DFT calculations confirm this assumption as they also reveal that Cu top positions are the most attractive sites.

My impression is that the authors have good reasons for asserting the contrast as shown. Perhaps they leaked in CO, which is known to bond to top sites on all surfaces discussed and were able to orient their images with these marker molecules. They also (Ref. 40) have at least thought about this before but assert that it is still an open question.

Author reply: The assignment (i.e. dark features imaged with CO tips = Cu top sites) was already described by Schuler et al Ref 48. For symmetry reasons the dark feature were assigned to Cu top sites. This assignment has been used in the literature to determine adsorption positions of e.g. Fe atoms (Ref 52) or organic molecules (e.g. Ref. 25, 48-51). However, if the tip-surface distance during constant height scanning is too small the contrast can be ambiguous, which makes the assignment difficult (see e.g. Ref. 40) We believe it is very important to point this out as this method of determining adsorption positions is used often in the community for revealing reaction mechanisms via comparison with theoretical adsorption structures.. We amended the paragraph "This also solves caveats..." on page 4 for clarification.

Making the assignment via adsorbed CO can also be ambiguous as the tip may bend differently on the surface and the CO molecule. See paragraph "Moreover, our findings enable a precise..." on page 12 for a discussion and Fig. S9, which demonstrates this for an organic molecule.

The discussion of the DFT results are very important and I would ask the authors to lengthen this section. Currently, they state, "the experiments for Ag(111) revealed a negative force difference

[inverted contrast, Fig. 2(b)], while the DFT calculations give a slightly positive value. We rationalize that this is caused by the larger and more complex metal tip that is used in the experiments and/or the limits for DFT-D3 for describing such small force differences.". Two points: (1) How would a larger and more complex metal tip change the contrast? Are you arguing electrostatic interaction via the large metal dipole of the tip apex or that the structure of the metal tip changes the chemical binding of the CO at the apex? (2) What are the limits of DFT-D3? 2 pN? A better description would be useful for future investigations wanting to use it.

Author reply:

Regarding Point 1: We conducted additional tests using a larger metal tip composed of three layers - comprising one, three, and six atoms, with the six atoms fixed to represent the bulk. The results are in good agreement with those obtained using the smaller tip model (two layers comprising one apex atom and three constrained atoms), demonstrating that the interaction is dominated by the CO–surface interaction, particularly involving the oxygen atom. These findings further support the conclusions presented in the main text and are summarized in Figure S7 of the Supporting Information.

Regarding Point 2: This is indeed a challenging question, as there is no well-defined force accuracy limit for systems like ours. Weymuth et al. (<https://doi.org/10.1021/acs.jctc.8b00078>) report a mean absolute error (MAE) of 0.49 kcal/mol for the PBE-D3(BJ) approach, though this estimate is based on gas-phase molecules and is therefore not directly transferable to our surface–tip system. Therefore, it remains difficult to specify a precise force accuracy threshold within the context.

It should also be noted that in DFT studies, it is often the qualitative trends in energies (and in this case, in forces) that are analyzed, rather than the absolute values. This means that even if the calculated values would lie outside the formal accuracy limits of DFT, the trends are still often reproduced reliably. See, for example, references Beilstein J. Org. Chem. 2014, 10, 1775, ChemPhysChem 2025, 26, e202400865 or J. Comp. Chem. 2014, 35, 986 for supporting evidence.

In conclusion, we explicitly acknowledge the relevance of the reviewer’s concern. It is true that we are dealing with very small energy and force differences, which approach the limits of what can be reliably captured by DFT. Nevertheless, there is a remarkable agreement between our theoretical predictions and the experimental observations and trends. The fact that DFT is capable of reproducing these subtle variations and their trends in accordance with the experimental results highlights its high level of accuracy in capturing trends, even as we approach the inherent limitations of the method in modeling complex surface–tip interactions.

I would also note that the application to adsorbates in general requires that the adsorbate is also relatively inert and that the surface only acts as a potential energy landscape. For many catalytic molecules this is not the case, as surface interaction is what weakens bonds, etc.

Author reply: For clarification we amended the following sentence on page 3: “For example, using halogenated AFM tips would enable to obtain in-depth information about halogen-metal interactions that are responsible for the catalytic dehalogenation process during on-surface Ullmann-type coupling as they weaken the halogen-carbon bond in the precursors. Or in other words, determining the strength of halogen-metal interactions at different atomic sites via our

approach can give useful information about where and when a dehalogenation reaction of a molecular precursor might actually happen.”

I have some smaller comments that I recommend the authors consider when they have adequately modified the manuscript to address the major issue described above:

- Consider "ultra-weak" instead of "ultra-low", as low is sometimes used for a spatial coordinate

Author reply: We replaced “ultra-low” by “ultra-weak” and “ultra-small” throughout the text.

- (p. 2) The authors talk about image contrast inversion. However, this was not properly explained. vdW attraction would nominally make one expect attraction. To understand this assumption, one has to understand that your hypothesis is interaction dominated by Pauli repulsion, which is not stated.

Author reply: We added the following sentence on page 2: “In case the contrast is caused by net repulsive tip-surface interactions, which are usually dominated by Pauli repulsion and short range electrostatics, 55-56 a non-inverted contrast is expected.”

- To make the work attractive to a broader community (chemistry), I would recommend discussing the well-known reactivity row of $\text{Cu} > \text{Ag} > \text{Au}$.

Author reply: We added the following sentence on page 11: “Furthermore, this is in agreement with the well-known reactivity row of $\text{Cu} > \text{Ag} > \text{Au}$.”

- Taking CO bending into account (p. 2) has been done before by e.g. Gross et al. Science 2012; Scheuerer et al PRL 2019; Hofmann et al New J Phys 2022

Author reply: We added references to the works of Scheurer and Hofmann (see [63] and [66] on page 2). The paper by Gross et al was already cited in this context, see ref [32] on page 2.

I would ask that the authors incorporate a comment about their work in contrast to that published by Neel and Kröger, Nano Letters 2021, especially regarding the COOP analysis

Author reply: The aim of the work by Neel and Kröger is to describe the particular “dip-hump” shape of the force vs. distance curve by a lateral harmonic potential. We added a citation to this work along with the papers considering CO bending (see Ref. 67). An experimental or DFT analysis regarding the site specific chemical tip-surface interactions was however not performed in this work.

- (p. 3) The authors claim "a non-inverted contrast is observed" - it could be that contrast does not change and just shifts, as indicated by the yellow circles. This relates strongly to my main critique, above

Author reply: At the distance chosen in Fig. 1c the bright features are the most prominent features. We amended the sentence for clarification (see page 4): “After decreasing the distance by another 30 pm [Fig. 1(c)] a non-inverted contrast is observed in which the bright features are most pronounced.”

- Although I expect the answer to be no, I am curious as to whether any additional excitation was observed during measurements

Author reply: There is no atomic contrast visible in the dissipation channel of the images.

- Define x and z. This will also help the figures. Currently writing "distance" on both the x and y axes (e.g. Fig 2 d-f) should read "x" or "x distance"

Author reply: For clarification we now write “tip-surface distance z ”. Regarding the x or y direction with specified the crystallographic directions, such as (1-21) (1-10) with arrows in Fig 2 d-f and in Fig. 3.

- Vertical dashed lines were not clearly defined in Fig. 2

Author reply: Done.

- It's great that the authors checked their force-distance curves with several amplitudes. Because the curve has several inflection points, this is a nice check for ill-fitting as described in Sader et al Nat Nano 2018 - but this citation is missing.

Author reply: The following sentences and references were added to the methods section: “The amplitudes were calibrated using the constant current method described in Ref. 92. The reliability of the amplitude calibration was confirmed by measuring spectroscopy curves at different amplitudes (see Supporting Information Figure S1), which reveals that the resulting force vs distance curves are independent of the used amplitude calibration. 91-95”

- (p. 6) Hollow is not shown for Au or Ag in Fig. 2, so the following sentences are misleading: "Cu(111) and Ag(111) top sites (red) appear to be more attractive than hollow sites (orange) or bridge sites (blue). For Au(111), however, top sites are more repulsive than the other sites."

Author reply: The sentence was amended, now it reads: “For example, at the tip-surface distances marked by the black circles [which correspond to the relatively large imaging distances used in Fig. 2 (a-c)] the following differences between top sites (red), hollow sites (orange) and bridge sites (blue) are observed: For Cu(111) (Fig. 2g) top sites are more attractive than hollow and bridge sites. For Ag(111) (Fig 2h) top sites are more attractive than bridge sites. For Au(111) (Fig. 2i), however, top sites are more repulsive than bridge sites.”

- Is CO bending ignored in the discussion of Figs. 3 and 4? Is that valid?

Author reply: The 2D force and potential fields shown in Figure 3 were derived from the data in Fig. 2d,f where the bending of the CO is explained in detail (see white lines in Fig. 2d,f). Neglecting CO bending in the DFT calculations (see Fig. 4) is justified, as the distinction between the more attractive adsorption sites - top vs. bridge, and hcp vs. fcc – arises in a distance regime where the CO molecule remains linear. This is particularly evident when CO bending is artificially induced at short distances by slightly tilting the tip (see Supporting Information Fig. S6); at larger tip-sample distances, the resulting force–distance curves are indistinguishable from those obtained with a perfectly symmetric tip and an unbent CO configuration.

- The authors claim "By introducing a small asymmetry, we can reproduce the bending of the CO molecule and the shape of the experimental curves". This is very important and I would ask them to include a figure and discussion of this in the main text or supporting material

Author reply: We included Supporting Information Fig. S6, which presents force–distance curves calculated at the top, bridge, fcc, and hcp sites on Cu(111), Au(111), and Ag(111), using a slightly asymmetric tip configuration. In this set-up, the entire tip is tilted (pitched) slightly toward the ab plane, such that it is no longer perfectly perpendicular to the surface. At larger tip-surface distances, the resulting curves closely resemble those shown in Fig. 4. At shorter distances, however, bending of the CO molecule is observed. As CO bending only takes place at small distances, our overall conclusions remain unaffected.

- (p. 9) The energy window of -8 to 4 eV seems arbitrary - can you justify it?

Author reply: The energy range shown in Fig. 4 was selected to match that of Supporting Information Fig. S8, in order to highlight the bonding and anti-bonding contributions of the O(*s*) and O(*p*) orbitals, as well as the M(*s*), M(*p*), and M(*d*) orbitals.

Reviewer #4 (Remarks to the Author):

Tschakert et al. used low-temperature AFM with a CO-tip to investigate the bare (111) surfaces of Cu, Ag and Au. They propose that they can align the AFM images with the underlying lattice. This would be very helpful, as they describe, for any studies of molecular adsorbates on these surfaces. Moreover, with the relatively inert CO tip they effectively probe the potential energy landscape of the surface and show just how flat these densely-packed surfaces are (lateral diffusion barriers less than 1 meV). Given the wide use of these surfaces, I believe such results are of interest generally in the chemistry and surface science community.

My fundamental problem with the manuscript in its current state is that I do not understand the assignment of the AFM images to the underlying lattice. For example, we are presented with an AFM image in Figure 1a and told that the dark circles correlate with atomic positions. Why? There are also lighter protrusions that, from symmetry arguments, could equivalently be the atomic positions. I expected this to be clearly explained when the measurements of potential energy as a function of height (Fig. 3) were discussed: Perhaps the atomic positions themselves would be seen via Pauli repulsion as maxima in the energy positions. But the dashed lines are over an energy minimum at the lowest height. Finally I thought that the DFT would be used to explain this. But the DFT calculations show a different assertion than the authors assert for Ag. Therefore can we believe it for Cu and Au?

Author reply: We appreciate the valuable assessment of our manuscript. We also have another series of images of Cu(111) with the same CO-tip, which contains 41 images at different distances and took about 24 h. The images shown in the manuscript were measured in a separate series to ensure that the effect of thermal drift is as low as possible. In the supporting information we added images and spectroscopy curves from the other series. Please see Fig. S2d, which was taken at $z = -354$ pm (about 60 pm more far away than Fig. 1a in the manuscript). The first visible contrast that appears at rather large distances is only composed of (very faint) dark features. Hitherto the dark features were assigned to top sites only because of symmetry reasons. Our DFT calculations confirm this assumption as they also reveal that Cu top positions are the most attractive sites.

Thank you for your thorough analysis of drift and creep, however this is not what I meant.

(1) If one looks again at Fig. 1, one would notice that there are high-symmetry bright protrusions as well. Consider please the black circles that I have drawn and pay attention that they indeed correspond to high-symmetry positions at the highest (-411 pm) slice. Therefore I disagree with the symmetry argument:

(2) Given that you explicitly state on page 4 “Our results presented here show that i) the dark features that are observed in AFM frequency images of Cu(111) surface atoms with CO tips at rather large tip-surface distances correspond to Cu top sites”, I expected a reason as to why the dark sites were chosen.

However, as you mention in your reply to my next comment, the assignment is based upon previous literature. Therefore I recommend a citation to Schuler et al. (or maybe all your citations you listed in the next reply) in the sentence on page 4 where you state, “At the largest tip-surface separation [Fig. 1(a)], the Cu atoms (red circles) appear as dark regions (inverted contrast).” because (1) this is not clear by symmetry reasons and (2) this is (as you state) an assignment by literature.

My impression is that the authors have good reasons for asserting the contrast as shown. Perhaps they leaked in CO, which is known to bond to top sites on all surfaces discussed and were able to orient their images with these marker molecules. They also (Ref. 40) have at least thought about this before but assert that it is still an open question.

Author reply: The assignment (i.e. dark features imaged with CO tips = Cu top sites) was already described by Schuler et al Ref 48. For symmetry reasons the dark feature were assigned to Cu top sites. This assignment has been used in the literature to determine adsorption positions of e.g. Fe atoms (Ref 52) or organic molecules (e.g. Ref. 25, 48-51). However, if the tip-surface distance during constant height scanning is too small the contrast can be ambiguous, which makes the assignment difficult (see e.g. Ref. 40) We believe it is very important to point this out as this method of determining adsorption positions is used often in the community for revealing reaction mechanisms via comparison with theoretical adsorption structures.. We amended the paragraph “This also solves caveats...” on page 4 for clarification.

Making the assignment via adsorbed CO can also be ambiguous as the tip may bend differently on the surface and the CO molecule. See paragraph “Moreover, our findings enable a precise...” on page 12 for a discussion and Fig. S9, which demonstrates this for an organic molecule.

The discussion of the DFT results are very important and I would ask the authors to lengthen this section. Currently, they state, “the experiments for Ag(111) revealed a negative force difference [inverted contrast, Fig. 2(b)], while the DFT calculations give a slightly positive value. We rationalize that this is caused by the larger and more complex metal tip that is used in the experiments and/or the limits for DFT-D3 for describing such small force differences.”. Two points: (1) How would a larger and more complex metal tip change the contrast? Are you arguing electrostatic interaction via the large metal dipole of the tip apex or that the structure of the metal tip changes the chemical binding of the CO at the apex? (2) What are the limits of DFT-D3? 2 pN?

A better description would be useful for future investigations wanting to use it.

Author reply:

Regarding Point 1: We conducted additional tests using a larger metal tip composed of three layers - comprising one, three, and six atoms, with the six atoms fixed to represent the bulk. The results are in good agreement with those obtained using the smaller tip model (two layers comprising one apex atom and three constrained atoms), demonstrating that the interaction is dominated by the CO-surface interaction, particularly involving the oxygen atom. These findings further support the conclusions presented in the main text and are summarized in Figure S7 of the Supporting Information.

Wonderful – thank you.

Regarding Point 2: This is indeed a challenging question, as there is no well-defined force accuracy limit for systems like ours. Weymuth et al. (<https://doi.org/10.1021/acs.jctc.8b00078>) report a mean absolute error (MAE) of 0.49 kcal/mol for the PBE-D3(BJ) approach, though this estimate is based on gas-phase molecules and is therefore not directly transferable to our surface–tip system. Therefore, it remains difficult to specify a precise force accuracy threshold within the context.

It should also be noted that in DFT studies, it is often the qualitative trends in energies (and in this case, in forces) that are analyzed, rather than the absolute values. This means that even if the calculated values would lie outside the formal accuracy limits of DFT, the trends are still often reproduced reliably. See, for example, references Beilstein J. Org. Chem. 2014, 10, 1775, ChemPhysChem 2025, 26, e202400865 or J. Comp. Chem. 2014, 35, 986 for supporting evidence.

As you wrote in your supporting information, you allowed the atoms to relax to a convergence criterion of 0.1 meV. Wouldn't that be a reasonable limit of accuracy?

In conclusion, we explicitly acknowledge the relevance of the reviewer's concern. It is true that we are dealing with very small energy and force differences, which approach the limits of what can be reliably captured by DFT. Nevertheless, there is a remarkable agreement between our theoretical predictions and the experimental observations and trends. The fact that DFT is capable of reproducing these subtle variations and their trends in accordance with the experimental results highlights its high level of accuracy in capturing trends, even as we approach the inherent limitations of the method in modeling complex surface–tip interactions.

All other comments were replied to in an outstanding manner.